# Serum 25-Hydroxyvitamin D and Cancer Risk: A Systematic Review of Mendelian Randomization Studies

**DOI:** 10.3390/nu15020422

**Published:** 2023-01-13

**Authors:** Thomas Lawler, Shaneda Warren Andersen

**Affiliations:** 1Carbone Cancer Center, University of Wisconsin-Madison, Madison, WI 53726, USA; 2Department of Population Health Sciences, School of Medicine and Public Health, University of Wisconsin-Madison, Madison, WI 53726, USA

**Keywords:** vitamin D, Mendelian randomization, cancer, African-American

## Abstract

Epidemiological studies suggest that higher serum 25-hydroxyvitamin D is associated with lower risk for several cancers, including breast, prostate, colorectal, and lung cancers. To mitigate confounding, genetic instrumental variables (IVs) have been used to estimate causal associations between 25-hydroxivtamin D and cancer risk via Mendelian randomization (MR). We provide a systematic review of 31 MR studies concerning 25-hydroxyvitamin D and cancer incidence and mortality identified from biomedical databases. MR analyses were conducted almost exclusively in European-ancestry populations and identified no statistically significant associations between higher genetically predicted 25-hydroxyvitamin D and lower risk for total cancer or colorectal, breast, prostate, lung, or pancreatic cancers. In recent studies including ≥80 genetic IVs for 25-hydroxyvitamin D, null associations were reported for total cancer (odds ratio [95% confidence interval] per 1-standard deviation increase: 0.98 [0.93–1.04]), breast (1.00 [0.98–1.02]), colorectal (0.97 [0.88–1.07]), prostate (0.99 [0.98–1.01]), and lung cancer (1.00 [0.93–1.03]). A protective association was observed for ovarian cancer in the Ovarian Cancer Association Consortium (0.78 [0.63–0.96] per 20 nmol/L increase, *p*-trend = 0.03), but not in the UK Biobank (1.10 [0.80–1.51]). Null associations were reported for other tumor sites (bladder, endometrium, uterus, esophagus, oral cavity and pharynx, kidney, liver, thyroid, or neural cells). An inconsistent protective association for cancer-specific mortality was also observed. Results from MR analyses do not support causal associations between 25-hydroxyvitamin D and risk for cancer incidence or mortality. Studies including non-White populations may be valuable to understand low 25-hydroxyvitamin D as a modifiable risk factor in populations with a higher risk of common cancers, including African ancestry individuals.

## 1. Introduction

An inverse association between serum vitamin D levels (25-hydroxyvitamin D) and risk for cancer is commonly reported in epidemiological studies [1,2,3]. The 25-hydroxyvitamin D is the primary circulating form of vitamin D and the clinical marker. There is widespread interest in determining whether 25-hydroxyvitamin D is a modifiable risk factor for cancer incidence or mortality, given that 25-hydroxyvitamin D can be easily increased through sun exposure and dietary supplements [4]. A protective association has been most commonly reported for colorectal cancer, with meta-analyses of longitudinal studies suggesting a 30–40% reduced risk for colorectal cancer in the highest quantile of serum 25-hydroxyvitamin D, compared to the lowest quantile [5,6,7]. Likewise, protective associations have been reported for other common cancers, including breast and lung cancers, although these findings have been less consistent than observed for colorectal cancer [8,9,10,11,12,13].

A protective role for vitamin D in cancer is supported by a wide variety of evidence from cell culture and preclinical models [1,2,3]. In vitro, vitamin D can reduce cell proliferation by decreasing the expression of cyclins and cyclin-dependent kinases [14,15,16,17]. Vitamin D can increase apoptotic cell death [2,15], and decrease angiogenesis by reducing the expression of vascular endothelial growth factor [2,18,19]. Vitamin D also attenuates pro-inflammatory signaling by reducing the nuclear translocation of transcription factor Nuclear-Factor Kappa Beta [1,20,21] and decreasing the expression of cyclooxygenase-2 [22,23], an enzyme that drives tumor development through the synthesis of prostaglandins [24]. In the colon and rectum, vitamin D may increase the expression of “tight-junction” proteins and thereby prevent the translocation of pro-inflammatory bacterial metabolites across the intestinal epithelium, a potential protective mechanism in colorectal cancer [25]. Preclinical studies have shown that vitamin D treatment can reduce tumor growth and multiplicity in mouse models of spontaneous, diet-induced cancer, as well as tumor xenograft models, chemically induced carcinogenesis, and germline mutations that mimic human carcinogenesis, as reviewed previously [1,3].

Despite promising findings from epidemiological studies and preclinical models, the National Academy of Medicine concluded that there is insufficient evidence to support a causal relationship between vitamin D and cancer [26]. Although promising, the myriad epidemiological findings of protective associations may suffer from residual confounding by multiple cancer risk factors, including obesity, physical activity, smoking, alcohol consumption, and diet patterns [27]. Reverse causation is also a concern for epidemiologic studies of 25-hydroxyvitamin D and cancer risk, where individuals with undiagnosed cancer reduce physical activity and time spent outdoors, leading to lower 25-hydroxyvitamin D. Additionally, because the relevant protective exposure window for 25-hydroxyvitamin D in human carcinogenesis is not known, differences between observational studies concerning participant age and length of follow-up may obscure protective effects of 25-hydroxyvitamin D and create inconsistencies in the literature. While there have been several large-scale randomized controlled trials (RCTs) of vitamin D and incident cancer outcomes, these trials have reported mostly non-significant results [28,29,30]. In the nationwide Women’s Health Initiative (WHI) clinical trial, 36,282 postmenopausal women were randomized to receive calcium plus vitamin D supplements (1 g Ca/400 IU vitamin D-3) for seven years, with no effect on colorectal cancer [28] (hazard ratio (HR) 95% confidence interval [CI]] = 1.08 [0.86–1.34]), invasive breast cancer [31] (HR = 0.96 [0.85–1.09]), or total cancer incidence [32] (HR = 0.98 [0.90–1.05]). In the Vitamin D and Omega-3 Trials (VITAL), including 25,871 participants age 50+, 2000 IU/day vitamin D did not reduce the risk for colorectal cancer (HR = 1.09 [0.73–1.62]), breast cancer (HR = 1.02 [0.79–1.31]), prostate cancer (HR = 0.88 [0.72–1.07]), lung cancer (HR = 1.00 [0.73–1.38]), or invasive cancer of any type (HR = 0.96 [0.88–1.06]) over a median 5.3 years of follow-up [29,33], although a significantly reduced risk for combined metastatic cancer or cancer death was reported (HR = 0.83 [0.69–0.99]) [33]. Of particular interest, the VITAL study included oversampling of African Americans (who are more likely to be vitamin D deficient [34]), and a borderline protective effect for any invasive cancer was observed in this subgroup (HR = 0.77 [0.59–1.01]). Likewise, the Vitamin D Assessment Study (VIDA) showed no significant effect of high-dose vitamin D treatment (100,000 IU bolus monthly) on total cancer incidence over 2–4 years of follow-up (HR = 1.01 [0.81–1.25]) [30], and two recent meta-analyses of RCTs reported no significant effect of vitamin D treatment on risk for either total cancer (HR = 0.98 [0.93–1.03]) or invasive breast cancer incidence (HR = 1.04 [0.85–1.29]) [35,36]. 

However, these trials have been attended by several noteworthy limitations. Most studies have had relatively limited follow-up with participants (approximately 2–7 years), which may be insufficient for detecting protective effects on cancer outcomes that take more than 10 years to develop. Other problems have included high rates of participant drop-out and drop-in (i.e., control group participants who begin taking vitamin D supplements), which may cause systematic measurement error and difficulty interpreting results [28]. In a re-analysis of WHI data, there was evidence that calcium plus vitamin D significantly reduced the risk for total cancer (HR = 0.86 [0.78–0.96]) and invasive breast cancer (HR = 0.80 [0.66–0.96]) after excluding participants who consumed non-protocol calcium or vitamin D supplements at randomization [37], while a non-significant 17% reduced risk for colorectal cancer was also reported. These trials have also been limited in their inclusion of participants with vitamin D deficiency or insufficiency (i.e., <20 ng/mL per National Academy of Medicine guidelines [26]), a population that may be more likely to benefit from interventions to increase serum 25-hydroxyvitamin D levels.

Mendelian randomization (MR) studies can be used to draw causal inferences by utilizing common genetic variants as instrumental variables (IVs) [38]. In a MR study, genetic variants associated with an exposure of interest (e.g., serum 25-hydroxyvitamin D) are identified from a large-scale genome-wide association study (GWAS) and subsequently applied to an independent data set to derive an unbiased estimate of the exposure-outcome association [39] (see Figure 1). The MR study design offers several advantages over traditional epidemiological studies [38]. As exposure-related genotypes are randomly distributed upon conception, MR studies are less susceptible to bias by confounding, mirroring the randomization process from clinical trials [38]. Further, as genotypes are inherited at conception, MR studies reflect lifelong exposure to a nutrient/exposure of interest, and are therefore more likely to capture the relevant exposure period compared to traditional epi studies and clinical trials [40]. As a practical benefit, MR analysis can be implemented from existing data sets, and consequently are efficient to conduct [39,41]. In recent years, the number of MR studies reported in the literature has steadily increased, reflecting the greater availability of GWAS data for an ever-expanding number of traits, the emergence of large-scale consortia studies with enormous sample sizes, and online platforms for the analysis of publicly available data sets [41].

MR studies require several assumptions that must be satisfied for the analysis to be valid [42]. (1) Genetic IVs must be valid instruments, meaning that they are associated with the exposure of interest (e.g., serum 25-hydroxyvitamin D). This assumption is usually satisfied by utilizing IVs identified in large-scale GWAS with a high degree of statistical significance. Related to validity is the concept of “instrument strength”, which indicates the magnitude of the IV-exposure association and affects the statistical power for detecting causal associations with the outcome of interest [43]. (2) Genetic IVs must not be associated with known risk factors for the outcome(s) of interest (i.e., absence of confounding). This assumption can be tested by searching for IV–confounder associations in published GWAS, or by examining these associations in the outcome sample when individual-level data are available. Additionally, (3) MR analysis requires the strong assumption that associations between genetic IVs and the outcome(s) are mediated only by the exposure of interest, and consequently that there are no other causal pathways connecting the IVs to the outcome(s), also known as the absence of pleiotropy [42]. Although several statistical tools have been developed to investigate and accommodate pleiotropic effects of IVs (e.g., MR–Egger and weighted-median MR [44,45]), this assumption requires careful consideration and deep knowledge of the relevant biological pathways that contribute to the pathophysiology of the outcome(s). This assumption will generally become more problematic as the number of genetic IVs included in the analysis increases, reflecting the increased power of GWAS for detecting genotype–exposure associations [45].

Given the inconsistency of epidemiologic studies and to minimize the risk for confounding, MR analyses have been utilized to investigate the causal association between serum 25-hydroxyvitamin D and multiple cancer outcomes, including overall cancer, colorectal, breast, prostate, and lung cancer risk. These studies have been made possible by the publication of large-scale GWAS for 25-hydroxyvitamin D, which have increased the number of genotypes available for IV analysis [46,47,48]. The primary objective of this review is to comprehensively summarize the MR literature concerning 25-hydroxyvitamin D and incident cancer outcomes, including causal estimates and key study parameters including sample size and instrument strength. A secondary objective is to report causal estimates for the association with cancer-specific mortality, which has been investigated less frequently. In the discussion, we aim to highlight avenues for future investigation using MR, including the need to investigate 25-hydroxyvitamin D as a modifiable risk factor for cancer specifically in non-European ancestry populations.

## 2. Materials and Methods

### 2.1. Literature Review

We performed a systematic review utilizing PRISMA guidelines to identify all published MR studies where serum 25-hydroxyvitamin D was an exposure of interest and cancer or cancer mortality was included as an outcome. Incident cancer outcomes of interest included total invasive cancer and all organ-specific cancer types. The lead author identified eligible manuscripts through May 2022 utilizing online databases including Pubmed, Scopus, and Web of Science. Keywords utilized for search were “vitamin D” OR “25-hydroxyvitamin D”, AND “Mendelian randomization”, AND “cancer”, OR “carcinoma”, OR “adenoma” OR “neoplasms”, OR “leukemia”, OR “lymphoma”, OR “glioma”, OR “myeloma”, OR “melanoma”. No restrictions were placed on date of publication. After completion of the literature review, titles and abstracts were reviewed from all manuscripts to identify studies that did not meet the inclusion criteria. Subsequently, the full text from each remaining manuscript was reviewed to confirm eligibility for inclusion. For all studies that met inclusion criteria, reference lists were reviewed to identify additional publications. Papers that reported associations between vitamin D-related IVs and cancer outcomes without providing MR estimates were excluded. Literature review results are presented in Figure 2.

### 2.2. Data Extraction

Key information was extracted from each manuscript by the lead author, including the number of cases and controls for each cancer outcome, the racial ancestry of the sample, the number of single nucleotide polymorphisms (SNPs) used as genetic IVs, instrument strength (including percentage of variation explained (PVE) for 25-hydroxyvitamin D and F-statistic, where available), odds ratios (ORs) and/or risk ratios (RRs) from MR analysis (with 95% CI), and *p*-trend. Where necessary, abstracted ORs/RRs were inverted to reflect estimates for increasing serum 25-hydroxyvitamin D levels. A majority of manuscripts included a primary MR estimate calculated via inverse-variance weighting (IVW) of Wald ratios (obtained for each genetic IV), as well as multiple sensitivity analyses to account for potential pleiotropy, including MR–Egger [44], weighted-median MR [45], and/or MR-PRESSO [49]. Given the widespread agreement in results utilizing these different MR estimates, only the primary IVW MR estimates for each manuscript are presented here (i.e., IVW estimate for MR studies utilizing summary statistics only, and Wald-type ratio for one-sample MR studies utilizing individual-level data), and notable sensitivity analyses, such as those that affect statistical significance of MR estimates, are described in the text. Likewise, separate estimates for prominent cancer subtypes (e.g., ER+ vs. ER- breast cancer, distal vs. proximal colon cancer) were not reported, barring strong evidence for effect heterogeneity. If multiple MR estimates for the same cancer outcome were obtained using independent samples or different sets of genetic IVs, we reported each estimate separately. Strength of the genetic IVs (e.g., PVE and/or F-statistic) was not consistently reported across studies due to differences in study design and the availability of individual-level data in the outcome sample. When individual-level data were utilized to calculate PVE and/or F-statistic for the IV-25-hydroxyvitamin D association among participants in the MR analysis sample, these statistics were abstracted and reported in Table 1. For manuscripts that reported results from summary statistics MR, or results from individual-level analysis where the IV-25-hydroxyvitamin D association was not assessed in study participants, instrument strength statistics were obtained from the 25-hydroxyvitamin D GWAS(s) that was utilized as a source of summary statistics, or from comparable 25-hydroxyvitamin D MR studies that included the same genetic IVs. Additional details concerning the assessment of instrument strength are provided in Appendix A.

## 3. Results

### 3.1. Overview of MR Studies

In total, 31 manuscripts reported MR estimates of the association between serum 25-hydroxyvitamin D and any cancer outcome, including cancer mortality (see Appendix A). MR estimates of the causal association for 25-hydroxyvitamin D were identified for total cancer (5) and the following cancer types: breast (9), colorectal (10), endometrial (2), esophageal (2), glioma (2), kidney (2), lung (8), neuroblastoma (3), ovarian (6), pancreatic (5), prostate (11), skin (8, including basal cell carcinoma, squamous cell carcinoma, and melanoma), as well as bladder cancer, leukemia, hepatocellular carcinoma, lymphoid cancer, non-Hodgkin’s lymphoma, cancer of the oral cavity or pharynx, thyroid cancer, or uterine cancer (1 each). Four studies were identified concerning serum 25-hydroxyvitamin D and cancer mortality. MR estimates and sample characteristics are provided in Table 1. Additional study details concerning study design, parent GWAS, instrument strength, sensitivity analyses, and assessment of confounding are provided in Appendix A. MR analyses have been conducted overwhelmingly in samples of European ancestry. Earlier studies (completed prior to 2020) utilized 3–6 genetic IVs for 25-hydroxyvitamin D (PVE approximately 1–3%), while studies conducted after 2020 included a larger numbers of genetic IVs (10–110 IVs) identified from GWAS analysis of UK Biobank participants [47,48]. It has been estimated that GWAS significant SNPs from analyses of the UK Biobank explain approximately 5–10% of the variability in serum 25-hydroxyvitamin D [47,48]. A majority of studies (23 of 31) assessed the effect of pleiotropy on MR estimates by performing MR–Egger regression, weight-median MR, leave-one-out analyses, and/or additional sensitivity analyses. For studies utilizing individual-level data, the “absence of confounding” assumption was typically assessed by testing for associations between genetic IVs and cancer risk factors (e.g., age, sex, body mass index, smoking, alcohol consumption, and exercise (Appendix A).

### 3.2. Summary of MR Estimates for Cancer Incidence

MR analyses have shown no statistically significant associations between increasing genetically predicted serum 25-hydroxyvitamin D and lower risk for colorectal, breast, prostate, lung, or pancreatic cancer, or for total cancer (Table 1). To summarize, 5 out of 10 studies showed an inverse association (i.e., OR < 1.00) for colorectal cancer, 1 of 9 for breast cancer, 4 of 8 for lung cancer, 5 of 11 for prostate cancer, 2 of 5 for pancreatic cancer, and 2 of 5 for total cancer. In a recent analyses including a larger number of genetic IVs for 25-hydroxyvitamin D identified in the UK Biobank (12–91 genetic IVs), Ye et al., (2021) reported null associations between genetically predicted 1-standard deviation increase in 25-hydroxyvitamin D and risk for breast (OR [95% CI]: 1.00 [0.98–1.02]), colorectal (1.00 [0.99–1.02]), lung (1.00 [0.97–1.03]), prostate (0.99 [0.98–1.01]), and pancreatic cancers (0.92 [0.76–1.01]) [50]. Additionally, utilizing genetic IVs identified from the UK-Biobank, Jiang et al., (2021), Ong et al., (2021), and He et al., (2022) reported similar null associations for genetically predicted 25-hydroxyvitamin D and risk for breast, colorectal, lung, prostate, and pancreatic cancers [57,58,62]. Likewise, Yuan et al. reported null associations with risk for total cancer utilizing 115 genetic IVs from the UK Biobank (0.98 [0.93–1.04] per 1-standard deviation increase) [78].

A protective association with ovarian cancer was reported by Ong et al., (2021) [58] in MR analyses of the Ovarian Cancer Association Consortium (OCAC) case-control study (OR [95% CI]: 0.78 [0.63–0.96] per unit increase in log-transformed 25-hydroxyvitamin D, *p*-trend = 0.03), and in other manuscripts utilizing the OCAC data set [50,70]. However, a non-significant association between genetically predicted 25-hydroxyvitamin D and ovarian cancer was reported by Ong et al., (2018) [53] in the UK Biobank study (1.10 [0.80–1.51] per 20 nmol/L increase, *p*-trend ≥ 0.57) and by Dimitrakopoulou et al., (2017) [51] in the Follow-up of Ovarian Cancer Genetic Association and Interaction Studies (FOCI) consortium (1.12 [0.86–1.47] per 25 nmol/L increase). While Ong et al., (2021) [58] reported that higher genetically determined 25-hydroxyvitamin D was associated with greater odds of basal cell carcinoma (1.18 [1.05–1.33] per 20 nmol/L increase, *p*-trend = 0.01), this finding was no longer significant after adjusting for skin pigmentation and episodes of childhood sunburn (1.15 [0.99–1.32]). No significant associations were reported for other cancer sites, including bladder, endometrium, uterus, esophagus, oral cavity and pharynx, kidney, liver, thyroid, or neural cells.

### 3.3. Summary of MR Estimates for Cancer-Specific Mortality

Afzal et al., (2014) reported that higher genetically determined 25-hydroxyvitamin D was associated with lower risk for cancer-specific mortality in analyses of the Copenhagen City Heart Study, the Copenhagen General Population Study, and the Copenhagen Ischemic Heart Disease Study (Table 2) (OR [95% CI]: 0.70 [0.50–0.98] per 20 nmol/L increase) [79]. No association with cancer-specific mortality was reported from the Women’s Genome Health Study (0.98 [0.73–1.32] per 20 nmol/L increase) [52], the UK Biobank (0.97 [0.84–1.11] per 20 nmol/L increase) [53], or most recently in a collaborative analysis of 386,406 individuals from the UK Biobank, EPIC-CVD, the Copenhagen City Heart Study, or the Copenhagen General Population Study (0.98 [0.93–1.02] per 10 nmol/L increase) [80]. In this analysis, a protective trend for higher genetically predicted 25-hydroxyvitamin D and reduced risk for cancer mortality was observed for individuals who were vitamin D deficient (<25 nmol/L), but this association did not reach statistical significance (0.81 [0.65–1.02], *p*-trend = 0.09).

## 4. Discussion

To our knowledge, this is the first systematic review of MR studies concerning serum 25-hydroxyvitamin D and cancer outcomes. While protective associations between 25-hydroxyvitamin D and multiple cancer outcomes have been frequently reported in epidemiological studies [5,6,8,10,11], results from MR studies do not support a causal association with risk for cancer, with the possible exception of ovarian cancer (for which results are inconsistent [50,51,53,58,70,71]). Notably, consistently null associations have been reported for most cancers despite a substantial increase in the number of genetic IVs for 25-hydroxyvitamin D, and consequently are not likely to be explained by weak instrument bias. These results are largely in agreement with published results from RCTs [28,29,30,31,32,35,37], including the large-scale VITAL study [29] and together suggest that interventions to increase serum 25-hydroxyvitamin D with sunlight or supplements are unlikely to have a substantial impact on cancer risk. As the majority of MR analyses have been conducted in samples of European ancestry, additional MR studies including racial and ethnic minorities may provide novel insights concerning the role of vitamin D in cancer etiology in these populations.

When all assumptions for MR analysis are met [42], MR can provide an unbiased estimate of the causal relationship between the exposure and outcome(s) of interest [40]. However, multiple considerations prevent the confident extrapolation of results obtained in European-ancestry samples to other populations, including racial and ethnic minorities. For example, African Americans have a much higher rate of vitamin D inadequacy (<20 ng/mL) compared to White Americans of European descent (82.1% vs. 30.9%, respectively) [34], as well as higher risk for multiple common cancers [e.g., breast, prostate, and colorectal cancers] [81]. Consequently, African Americans may be more likely to benefit from interventions to maintain adequate vitamin D levels compared to populations with higher prevalence of vitamin D adequacy. Importantly, the causal relationship between genetically determined serum 25-hydroxyvitamin D and risk for most cancers has not been investigated in African Americans, or other African-ancestry populations.

Likewise, another important consideration is that genetic IVs for serum 25-hydroxyvitamin D have been identified and validated in large GWAS of White, European-ancestry individuals (e.g., the UK Biobank [47,48] and the SUNLIGHT consortium [46]), and it is not clear that these instruments are valid for individuals of African ancestry. African-ancestry populations have a different haplotype structure compared to European populations, characterized by smaller regions of the genome in linkage-disequilibrium [82]. Consequently, vitamin D-related tag-SNPs identified in Europeans may not demonstrate the same associations with 25-hydroxyvitamin D in African-ancestry populations, even if the underlying causal variants are the same [82]. This would amount to a violation of the first and most basic MR assumption requiring the validity of IVs [42]. To calculate reliable MR estimates of the causal association between 25-hydroxyvitamin D and cancer in African-ancestry populations, it will first be necessary to identify ancestry-specific IVs by performing large-scale GWAS of African-ancestry individuals, or by statistically fine-mapping loci linked to 25-hydroxyvitamin D in European-ancestry samples to identify underlying causal variants [83].

An additional consideration in the interpretation of MR studies is statistical power. MR requires large numbers of cases for analysis and often has limited power to detect causal associations, as GWAS significant SNPs used as genetic IVs are likely to explain a small percentage of variability in the exposure of interest [43]. ‘Weak instrument bias’, generally defined as F-statistic < 10, tends to bias MR effect estimates towards the null, and hence obscures causal associations [43]. Initial GWAS studies identified only four independent loci associated with 25-hydroxyvitamin D, which mapped to genes implicated in vitamin D metabolism and transport (i.e., *GC*, *CYP2R1*, *DHCR7* and *CYP24A1*) [84,85]. In combination, it has been reported that these four variants explain approximately 1.0–2.5% of the serum 25-hydroxyvitamin D variability [76,79,86,87] (although considerably higher PVE was reported by Hiraki et al. [88]). More recently in the large-scale SUNLIGHT consortium (featuring 79,366 participants from 31 cohorts), two additional genome-wide significant loci were identified from *SEC23A* and *AMDHD1* and corroborated in an independent data set [46], and the six GWAS significant SNPs explained 2.84% of the variability in serum 25-hydroxyvitamin D. Consequently, most MR studies completed prior to 2020 included only 4–6 genetic IVs and may have been underpowered to detect causal effects. However, in independent analyses featuring more than 400,000 White British individuals from the UK Biobank, Revez et al. [47] and Manousaki et al., (2020) [48] reported 143 and 138 conditionally independent variants associated with 25-hydroxyvitamin D, respectively, including a significant number of rare variants (minor allele frequency < 0.05), and it was estimated that these variants explained approximately 5–10% of the variability in serum 25-hydroxyvitamin D [47,48]. Consequently, recent MR studies utilizing these additional genetic IVs were better powered to detect meaningful reductions in risk for common cancers with higher serum 25-hydroxyvitamin D [50,57,58,62]. For example, Ong et al. reported at least 90% power to detect a modest reduction in risk for melanoma and breast, prostate, and lung cancers (OR = 0.80 per one standard deviation increase) [58], while He et al. reported 80% power to detect modestly reduced risk for colorectal cancer (OR = 0.91 per standard deviation increase) [62]. Despite achieving greater statistical power, these recent MR studies have reported non-significant associations with multiple common cancers (including colorectal, breast, prostate, endometrial, and lung cancers, amongst others, with inconsistent results for ovarian cancer), and thus provide stronger evidence against a causal, protective effect for vitamin D against these cancers. 

However, it is likely that additional genetic IVs for 25-hydroxyvitamin D will be discovered as GWAS sample sizes continue to grow, increasing power for novel MR analyses. In a recent analysis from the UK Biobank, GWAS significant SNPs (n = 138) explained 4.9% of 25-hydroxyvitamin D variability, while the SNP-based heritability (h^2^) estimated from the same sample was 16.1% [48], indicating missing heritability that may be explained in part by rare variants or common variants with small effect sizes. As additional genetic IVs are discovered, greater statistical power may support the implementation of MR analyses not previously attempted including (1) identifying protective causal effects against rare cancers; (2) identifying protective effects of 25-hydroxyvitamin D in racial/ethnic minorities at heightened risk for common cancers and vitamin D deficiency (e.g., African Americans); and (3) investigating effect modification by established cancer risk factors [e.g., smoking, obesity, or family history of cancer]. Therefore, although published MR studies have shown consistent evidence against a causal effect of 25-hydroxyvitamin D in common cancers in White, European-ancestry populations, the thoughtful use of genetic instruments may still provide valuable insights concerning the role of vitamin D in human carcinogenesis.

Stronger instruments and enhanced statistical power will also facilitate additional MR analyses of serum 25-hydroxyvitamin D and cancer-specific mortality, for which there are currently sparse data. While Afzal et al., (2014) reported a significantly reduced risk for cancer-specific mortality for individuals with higher genetically predicted 25-hydroxyvitamin D (OR [95% CI]: 0.70 [0.50–0.98]) [79], other studies did not replicate this finding [52,53,80]. There is some difficulty in interpreting these results, as estimates of the effect of 25-hydroxyvitamin D on cancer mortality may have been biased by effects on cancer incidence. Results from experimental studies suggest that vitamin D may reduce risk for cancer-specific mortality through multiple mechanisms leading to the inhibition of tumor cell proliferation, invasiveness, and metastasis [2,14,22,89,90]. In support of this, a recent meta-analysis of five clinical trials (including 815 total colorectal cancer cases) demonstrated that supplementation with vitamin D-3 (400–4000 IU/day) may modestly reduce risk for colorectal cancer-specific mortality (HR [95% CI]: 0.70 [0.48–0.93]) [91]. While adequately powered clinical trials can provide stronger evidence to support short-term interventions with vitamin D to reduce cancer-specific mortality, MR studies incorporating a larger number of ancestry-specific genetic IVs may clarify whether greater lifelong exposure to vitamin D can help to mitigate the risk for cancer death.

In conclusion, despite plausible biological mechanisms and considerable supportive evidence from traditional epidemiologic studies, results from MR analyses do not support the existence of a causal association between serum 25-hydroxyvitamin D and risk for cancer. As the vast majority of relevant studies have been conducted in samples of European ancestry, MR studies of racial and ethnic minorities at higher risk for vitamin D deficiency and common cancers (e.g., African Americans), utilizing ancestry-specific genetic instruments should be prioritized.

## Figures and Tables

**Figure 1 nutrients-15-00422-f001:**
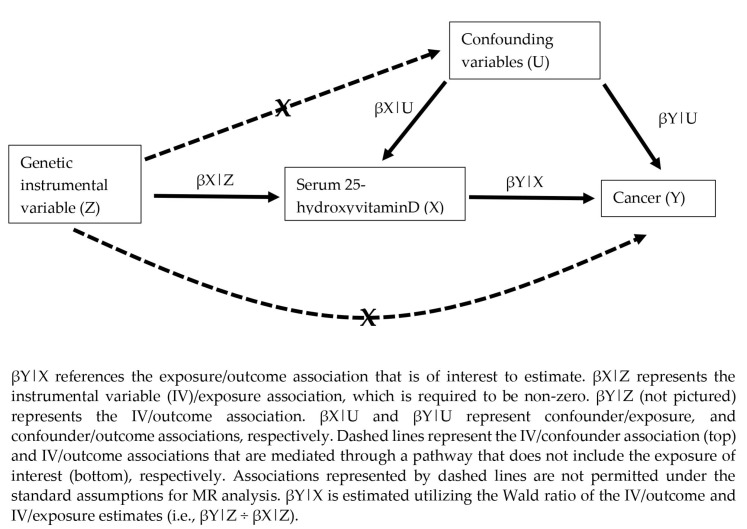
Overview of assumptions for Mendelian randomization (MR) studies.

**Figure 2 nutrients-15-00422-f002:**
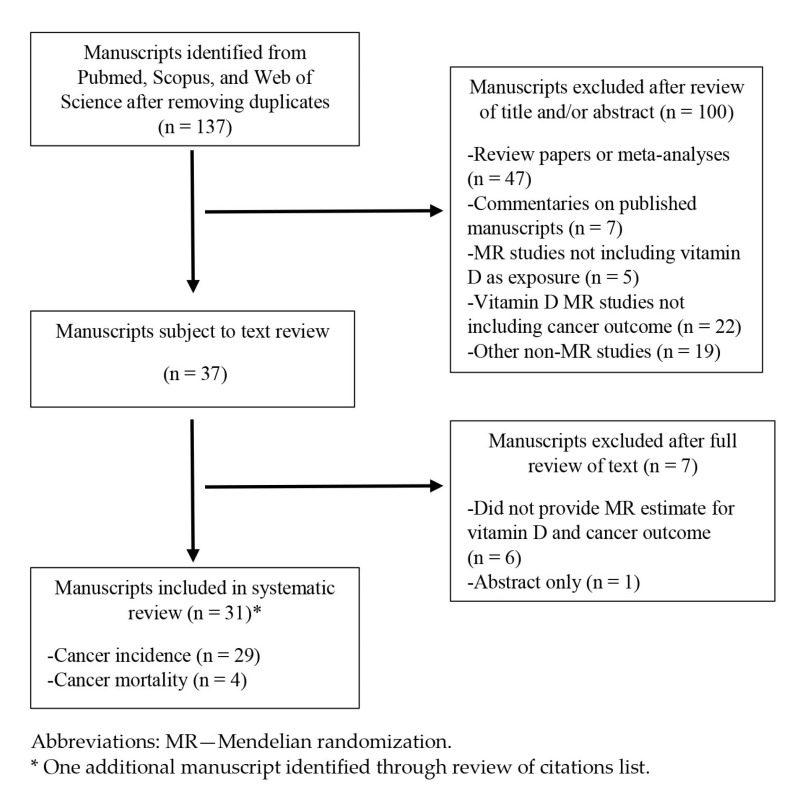
Flow diagram of studies included in systematic review.

**Table 1 nutrients-15-00422-t001:** Summary of all published Mendelian randomization (MR) estimates for the association between serum 25-hydroxyvitamin D and cancer outcomes. Additional study details are displayed in Appendix A.

Author, Year	Primary Ancestry	Cases	Controls	IV SNPs	Instrument Strength (PVE) ^a^	Instrument Strength (F-Statistic)	OR (95% CI)	Per Unit Change	*p*-Trend
	Bladder cancer								
Ye, 2021 [50]	European	NR	NR	61	7.5%	NR	1.00 (0.99–1.02)	1 SD increase	0.58
	Breast cancer								
Dimitrakopoulou, 2017 [51]	European	15,748	18,084	4	1.0–2.5%	NR	1.05 (0.89–1.24)	25 nmol/L increase	0.59
Chandler, 2018 [52]	European	1560	N/A ^b^	5	2.6%	48	1.14 (0.92–1.41)	20 nmol/L increase	0.22
Ong, 2018 [53]	European	11,703	NR	5	3.6%	NR	0.94 (0.85–1.03)	20 nmol/L increase	0.19
Wang, 2018 [54]	African	1657	2029	4	1.0–2.5%	NR	1.04 (0.97–1.11)	1 SD increase	0.23
Jiang, 2019 [55]	European	122,977	105,974	6	2.8%	NR	1.02 (0.78–1.08)	25 nmol/L increase	0.47
Cheng, 2020 [56]	European	1145	1142	3	NR	NR	1.09 (0.55–2.15)	1 SD increase	0.80
Jiang, 2021 [57]	European	122,977	105,974	88	4.9%	NR	1.02 (0.97–1.07)	1 SD increase	0.51
Ong, 2021 [58]	European	122,977	105,974	74	3.9%	NR	1.03 (0.93–1.13)	25 nmol/L increase	0.60
Ye, 2021 [50]	European	122,977	105,974	91	7.5%	NR	1.00 (0.98–1.02)	1 SD increase	0.95
	Colorectal cancer								
Theodoratou, 2012 [59]	European	2001	2237	4	1.0–2.5%	16.52	1.16 (0.60–2.23)	NR	>0.05
Dimitrakopoulou, 2017 [51]	European	11,488	11,679	4	1.0–2.5%	NR	0.92 (0.67–1.10)	25 nmol/L increase	0.36
	European	5100	4831	4	1.0–2.5%	NR	1.04 (0.78–1.38)	25 nmol/L increase	0.81
Chandler, 2018 [52]	European	329	N/A ^b^	5	2.6%	48	1.54 (0.96–2.47)	20 nmol/L increase	0.07
He, 2018 [60]	European	9940	22,848	6	2.8%	46.0	1.03 (0.51–2.07)	1 unit increase ^c^	0.93
	European	17,716	40,095	6	2.8%	46.0	0.91 (0.69–1.19)	1 unit increase ^c^	0.48
Ong, 2018 [53]	European	4442	NR	5	3.6%	NR	0.94 (0.79–1.13)	20 nmol/L increase	0.52
Cheng, 2020 [56]	Japanese	6692	27,178	7	NR	NR	1.01 (0.99–1.03)	1 SD increase	0.42
Cornish, 2020 [61]	European	26,397	41,481	5	2.6%	431.37	0.99 (0.90–1.09)	1 SD increase	0.89
Ye, 2021 [50]	European	NR	NR	61	7.5%	NR	1.00 (0.99–1.02)	1 SD increase	0.71
He, 2022 [62]	European	26,397	41,181	110	7.5%	25,241	0.97 (0.88–1.07)	1 unit increase ^c^	0.57
	Endometrial cancer								
Ong, 2018 [53]	European	1938	NR	5	3.6%	NR	0.90 (0.72–1.13)	20 nmol/L increase	0.38
Ong, 2021 [58]	European	12,906	108,979	75	3.9%	NR	0.93 (0.80–1.07)	20 nmol/L increase	0.32
	Esophageal cancer								
Dong, 2019 [63]	European	4112	17,159	6	2.8%	NR	0.68 (0.39–1.19)	20 nmol/L increase	0.18
Ong, 2021 [58]	European	4112	17,159	76	3.9%	NR	0.97 (0.78–1.20)	20 nmol/L increase	0.76
	Glioma								
Takahashi, 2018 [64]	European	12,488	18,169	4	1.0–2.5%	12.57	1.21 (0.90–1.62)	NR	0.20
Saunders, 2020 [65]	European	12,488	18,169	5	2.7%	431.37	0.99 (0.86–1.15)	1 SD increase	0.93
	Kidney cancer								
Ong, 2018 [53]	European	1012	NR	5	3.6%	NR	1.21 (0.84–1.76)	20 nmol/L increase	0.31
Ye, 2021 [50]	European	NR	NR	62	7.5%	NR	1.00 (0.99–1.01)	1 SD increase	0.96
	Leukemia								
Ye, 2021 [50]	European	NR	NR	57	7.5%	NR	1.01 (1.00–1.03)	1 SD increase	0.10
	Liver cancer (hepatocellular carcinoma)								
Liu, 2020 [66]	Chinese	721	2890	6	2.8%	NR	1.03 (0.31–3.47)	NR	>0.05
	Lung cancer								
Dimitrakopoulou, 2017 [51]	European	12,537	17,285	4	1.0–2.5%	NR	1.03 (0.87 to 1.23)	25 nmol/L increase	0.72
Chandler, 2018 [52]	European	330	N/A ^b^	5	2.6%	48	0.96 (0.55–1.68)	20 nmol/L increase	0.89
Ong, 2018 [53]	European	1863	NR	5	3.6%	NR	1.04 (0.83–1.30)	20 nmol/L increase	0.73
Sun, 2018 [67] One-sample MR	European	676	N/A ^b^	3	3.4%	197	0.96 (0.54–1.69)	25 nmol/L increase	0.88
Sun, 2018 [67] Two-sample MR	European	676	N/A ^b^	3	1.0–2.5%	197	0.99 (0.88–1.12)	10% increase	0.85
Jiang, 2021 [57]	European	11,348	15,861	81	4.9%	NR	1.13 (0.98–1.32)	1 SD increase	0.10
Ong, 2021 [58]	European	11,348	15,861	65	3.9%	NR	0.94 (0.78–1.13)	25 nmol/L increase	0.50
Ye, 2021 [50]	European	NR	NR	82	7.5%	NR	1.00 (0.97–1.03)	1 SD increase	0.84
	Lymphoid cancer								
Ong, 2018 [53]	European	3576	NR	5	3.6%	NR	1.10 (0.92–1.31)	20 nmol/L increase	0.29
	Multiple myeloma								
Went, 2020 [68]	European	7717	29,304	5	2.7%	431.37	1.08 (0.93–1.26) ^c^	1 SD increase	>0.05
	Neuroblastoma								
Dimitrakopoulou, 2017 [51]	European	1627	3254	4	1.0–2.5%	NR	0.76 (0.47–1.21)	25 nmol/L increase	0.24
Ong, 2021 [58]	European	1627	3254	26	3.9%	NR	0.74 (0.42–1.29)	25 nmol/L increase	0.29
Ye, 2021 [50]	European	1627	3254	10	7.5%	NR	0.92 (0.63–1.34)	1 SD increase	0.67
	Non-Hodgkin’s lymphoma								
Ye, 2021 [50]	European	NR	NR	60	7.5%	NR	1.00 (0.98–1.03)	1 SD increase	0.87
	Cancer of the oral cavity and pharynx								
Dudding, 2018 [69]	European	5133	5984	5	2.0–3.5%	NR	1.01 (0.74–1.40)	1 SD increase	0.93
	European	585	336,523	5	2.0–3.5%	NR	0.86 (0.58–1.27)	1 SD increase	0.44
	Ovarian cancer								
Ong, 2016 [70]	European	10,065	21,654	3	1.3%	NR	0.79 (0.66–0.94)	20 nmol/L increase	<0.05
Dimitrakopoulou, 2017 [51]	European	4369	9123	4	1.0–2.5%	NR	1.12 (0.86–1.47)	25 nmol/L increase	0.40
Ong, 2018 [53]	European	1031	NR	5	3.6%	NR	1.10 (0.80–1.51)	20 nmol/L increase	0.57
Yarmolinsky, 2019 [71]	European	25,509	40,941	5	2.6%	423	1.02 (0.72–1.44)	1 unit increase ^c^	0.93
Ong, 2021 [58]	European	25,509	40,941	76	3.9%	NR	0.78 (0.63–0.96)	1 unit increase ^c^	0.03
Ye, 2021 [50]	Unclear	18,174	26,134	104	7.5%	NR	0.96 (0.93–0.99)	1 SD increase	0.02
	Pancreatic cancer								
Dimitrakopoulou, 2017 [51]	European	1896	1939	4	1.0–2.5%	NR	1.36 (0.81–2.27)	25 nmol/L increase	0.25
Ong, 2018 [53]	European	500	NR	5	3.6%	NR	1.09 (0.63–1.88)	20 nmol/L increase	0.76
Lu, 2020 [72]	European	8769	7055	6	2.8%	NR	1.13 (0.71–1.80)	1 unit increase ^c^	0.60
Ong, 2021 [58]	European	1896	1939	27	3.9%	NR	0.93 (0.46–1.92)	25 nmol/L increase	0.99
Ye, 2021 [50]	European	3851	3934	12	7.5%	NR	0.92 (0.76–1.11)	1 SD increase	0.37
	Prostate cancer								
Dimitrakopoulou, 2017 [51]	Unclear	22,898	23,054	4	1.0–2.5%	NR	0.89 (0.77–1.02)	25 nmol/L increase	0.08
	European	14,159	12,712	4	1.9%	NR	1.08 (0.88–1.33)	25 nmol/L increase	0.47
Ong, 2018 [53]	European	7532	NR	5	3.6%	NR	0.91 (0.80–1.05)	20 nmol/L increase	0.19
Jiang, 2019 [55]	European	79,148	61,106	6	2.8%	NR	1.00 (0.93–1.07)	25 nmol/L increase	0.99
Cheng, 2020 [56]	European	NR	NR	8	NR	NR	1.00 (0.99–1.00)	1 SD increase	0.37
Kazmi, 2020 [73]	European	15,167	58,308	4	2.4%	253.15	1.00 (0.97–1.03)	1 SD increase	0.90
Zhang, 2020 [74]	European	4600	2941	3	1.0–2.5%	NR	1.16 (0.86–1.57)	NR	0.34
Jiang, 2021 [57]	European	79,194	61,112	51	4.9%	NR	0.98 (0.91–1.05)	1 SD increase	0.57
Ong, 2021 [58]	European	79,148	61,106	75	3.9%	NR	1.07 (0.89–1.29)	25 nmol/L increase	0.46
Ye, 2021 [50]	European	79,194	61,112	78	7.5%	NR	0.99 (0.98–1.01)	1 SD increase	0.42
Gu, 2022 [75]	European	51,704	227,795	138	8.2%	286.33	0.999 (0.995–1.003)	NR	0.72
	Skin cancer (non-Melanoma)								
Winsløw, 2018 [76]	European	8643	N/A ^b^	4	1.0%	314	1.11 (0.91–1.35)	20 nmol/L increase	>0.05
	Skin cancer (squamous cell carcinoma)								
Cheng, 2020 [56]	European	NR	NR	8	NR	NR	1.00 (0.99–1.00)	1 SD increase	0.47
Ong, 2021 [58]	European	7400	285,355	77	3.9%	NR	1.02 (0.88–1.19)	20 nmol/L increase	0.77
	Skin cancer (basal cell carcinoma)								
Ong, 2021 [58]	European	14,940	279,049	77	3.9%	NR	1.18 (1.05–1.33) ^d^	20 nmol/L increase	0.01
	Skin cancer (not specified)								
Ye, 2021 [50]	European	NR	NR	52	7.5%	NR	1.02 (0.99–1.04)	1 SD increase	0.15
	Skin cancer (melanoma)								
Ong, 2018 [53]	European	2758	NR	5	3.6%	NR	0.88 (0.71–1.10)	20 nmol/L increase	0.26
Cheng, 2020 [56]	European	NR	NR	8	NR	NR	1.00 (0.99–1.00)	1 SD increase	0.56
Liyanage, 2020 [77]	European	12,874	23,203	5	3.6%	NR	0.94 (0.84–1.05)	20 nmol/L increase	>0.05
Ong, 2021 [58]	European	15,990	26,409	69	3.9%	NR	1.09 (0.92–1.28)	20 nmol/L increase	0.31
	Thyroid cancer								
Ye, 2021 [50]	European	NR	NR	55	7.5%	NR	0.99 (0.96–1.02)	1 SD increase	0.56
	Total cancer								
Chandler, 2018 [52]	European	3985	N/A ^b^	5	2.6%	48	1.10 (0.96–1.25)	20 nmol/L increase	0.17
Ong, 2018 [53]	European	46,155	264,638	5	3.6%	NR	0.97 (0.90–1.04)	20 nmol/L increase	0.40
Ye, 2021 [50]	European	NR	NR	54	7.5%	NR	1.01 (1.00–1.02)	1 SD increase	0.19
Yuan, 2021 [78]	European	38,036	180,756	7	3.7% ^e^	NR ^f^	1.01 (0.97–1.05)	1 SD increase	0.68
	European	38,036	180,756	115	7.5%	NR ^f^	0.98 (0.93–1.04)	1 SD increase	0.50
	Uterine cancer								
Ye, 2021 [50]	European	NR	NR	59	7.5%	NR	1.01 (0.99–1.03)	1 SD increase	0.30

Abbreviations: SD—standard deviation; CI—confidence interval; GWAS—genome-wide association study; IV—instrumental variable; N/A—not applicable; NR—not reported; PVE—percentage of variation explained; SNP—single nucleotide polymorphism. ^a^ Additional details concerning calculation of PVE are provided in Appendix A. PVE represents either the percentage of serum 25-hydroxyvitamin D variance explained in a subgroup of the analysis sample, or the percentage of variance explained in the parent GWAS study. ^b^ Prospective cohort study design. ^c^ Serum 25-hydroxyvitamin D natural-log transformed to achieve normality. ^d^ The effect estimate for basal cell carcinoma was no longer significant after adjusting for genetic factors underlying skin pigmentation and childhood sunburn (OR [95% CI]: 1.15 [0.99–1.32]). ^e^ PVE estimates presented here differ from those presented in Yuan, 2021 Table 1 (see Appendix A for further explanation). ^f^ F-statistics reported in Yuan, 2021 Table 1 do not apply to MR analysis of cancer outcomes.

**Table 2 nutrients-15-00422-t002:** Summary of all published Mendelian randomization (MR) estimates for the association between serum 25-hydroxyvitamin D and cancer-specific mortality. Additional study details are displayed in Appendix A.

Author, Year	Primary Ancestry	Study Type	Sample Size	Cancer Deaths	IV SNPs	Instrument Strength (PVE) ^a^	Instrument Strength (F-Statistic)	OR (95% CI)	Per Unit Change	*p*-Trend
Afzal, 2014 [79]	European	Prospective cohort	95,766	2839	4	1.0%	NR	0.70 (0.50–0.98)	20 nmol/L increase	<0.05
Chandler, 2018 [52]	European	Prospective cohort	23,394	770	5	2.6%	48	0.98 (0.73–1.32)	20 nmol/L increase	0.90
Ong, 2018 [53]	European	Case-control	277,340 ^b^	6998	5	3.6%	NR	0.97 (0.84–1.11)	20 nmol/L increase	>0.05
Sofianopoulou, 2021 [80]	European	Prospective cohort	386,406	12,804	3–21 ^c^	1.8–5.8% ^c^	NR	0.98 (0.93–1.02)	10 nmol/L increase	0.29

Abbreviations: CI—confidence interval; IV—instrumental variable; NR—not reported; PVE—percentage of variation explained; SD—standard deviation; SNP—single nucleotide polymorphism. ^a^ Additional details concerning calculation of PVE are provided in Appendix A. PVE represents either the percentage of variance explained in a subgroup of the analysis sample, or the percentage of variance explained in the parent GWAS study. ^b^ Includes 270,342 controls. ^c^ The number of genetic IVs varied by cohort (see Appendix A for details).

## Data Availability

No new data were created or analyzed in this study. Data sharing is not applicable to this article.

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
