# Peer review of "Serum 25-Hydroxyvitamin D and Cancer Risk: A Systematic Review of Mendelian Randomization Studies"

_nutrients, 2023, doi:10.3390/nu15020422_

Round 1

Reviewer 1 Report

I am grateful for the opportunity to review this article. First of all, I would like to congratulate the authors for the enormity of work put into the preparation of this publication. The work presents an innovative and more precise approach to the long-explored topic of the role of vitamin D in the occurrence of cancer and related mortality. In my opinion, the only limitation of this study is the commonness of the topic of vitamin D and the overestimation of its therapeutic possibilities. Nevertheless, the use of an innovative method, which is Mendelian randomization, makes this article interesting and worth to publish.

Reviewer 2 Report

The review by Lawler et al. systematically summarized the evidence on vitamin D and cancer incidence and mortality from Mendelian randomization (MR) studies. The manuscript is well-written. I have only a few suggestions to help improve the clarity of the paper.

1.     As the authors mentioned, a valid MR study is based on no violation of the three core assumptions of instrumental variable analysis. Although we cannot completely confirm the other assumptions except for the relevance assumption, a thorough evaluation of the three assumptions for each original MR study is preferred.

2.     It is not appropriate to present the range of effect size for each outcome (Line 18-20 and Section 3.2) because they are not comparable due to different scales of the exposure used, e.g., per SD (absolute or log-transformed), per unit (absolute or log-transformed), per 25 nmol/L. Instead, the number of papers with inverse/positive association vs. the total number of papers on each outcome is recommended.

3.     Both one-sample and two-sample MR studies were included in this systematic review. IVW method is commonly used in two-sample MR analyses using summary-level data. However, the Wald-type ratio or two-stage MR is usually adopted in one-sample MR analyses using individual-level data. Thus, the statement “only the primary IVW MR estimates for each manuscript are presented here” (Line 193-194) could be problematic.

4.     The number of papers in the flowchart (Figure 2) is incorrect. A total of 100 papers was excluded after reviewing of title and/or abstract. However, the sum of the number of papers excluded for each reason is not equal to 100 (i.e., 47+7+5+22=81).
